

# GEB v0.1: A large-scale agent-based socio-hydrological model – simulating 10 million individual farming households in a fully distributed hydrological model

Jens A. de Bruijn[1,2], Mikhail Smilovic[1], Peter Burek[1], Luca Guillaumot[1], Yoshihide Wada[1,3], Jeroen C.J.H. Aerts[2]

[1]International Institute for Applied Systems Analysis (IIASA), Laxenburg, Austria
[2]Institute for Environmental Studies, VU University, De Boelelaan 1087, 1081 HV Amsterdam, the Netherlands
[3]Department of Physical Geography, Utrecht University, Utrecht, the Netherlands

*Correspondence to*: Jens A. de Bruijn (jens.de.bruijn@vu.nl)

**Abstract.** Humans play a large role in the hydrological system; for example, by extracting large amounts of water for irrigation, often resulting in water stress and ecosystem degradation. By implementing large-scale adaptation measures, such as the construction of irrigation reservoirs, water stress and ecosystem degradation can be reduced. Yet we know that many decisions, such as the adoption of more effective irrigation techniques or changing crop types, are made at the farm level by a heterogeneous farmer population. While these decisions are often advantageous for an individual farmer or their community, detrimental effects are frequently experienced downstream. Therefore, to fully comprehend how the human-natural water system evolves over time and space, and to explore which interventions are suitable to reduce water stress, it is important to consider human behaviour and feedbacks to the hydrological system simultaneously at the local household and large basin scales. Therefore, we present the Geographical, Environmental and Behavioural model (GEB), a coupled agent-based hydrological model that simulates the behaviour and daily bi-directional interaction of up to ~10 million individual farm households with the hydrological system on a personal laptop. GEB is dynamically linked with the spatially distributed grid-based hydrological model CWatM at 30'' resolution (< 1km at the equator). Because many small-holder farmer fields are much smaller than 1×1 km, CWatM was specifically adapted to implement dynamically sized hydrological response units (HRUs) at the farm level, providing each agent with an independently operated hydrological environment. While the model could be applied globally, we explore its implementation in the heavily managed Krishna basin in India, which encompasses ~8% of India's land area and ~11.1 million farmers. Here, we show how six combinations of storylines with endogenous and exogenous drivers of adaptation affect both the hydrological system and the farmer population.

## 1 Introduction

Water stress, defined as water demand exceeding water availability, is an increasing threat to human livelihood through, for example, decreasing agricultural yields, insufficient water for drinking and sanitation, and degrading ecosystems (Ablo and Yekple, 2018; van Leeuwen et al., 2016; Porporato et al., 2001; Kummu et al., 2016). A growing number of regions are expected to experience severe water stress in the future, largely driven by an increasing population and climate change




(Veldkamp et al., 2015; Kummu et al., 2016). Effective water management can help to reduce water stress but requires knowledge of the current status of water resources, socio-economic development, climate change, and the effects of interventions (Ibisch et al., 2016) on upstream and downstream water availability (Veldkamp et al., 2017). Therefore,

hydrological models, which simulate the hydrological system, are a widely used tool to provide an integrative vision of the system and formulate effective policies.

Humans play a large role in the hydrological system. For example, governments and other organizations construct reservoirs (Biggs et al., 2007) and channels for inter-basin transfers (Gupta and van der Zaag, 2008), disrupting natural flows. Small-scale adaptations, such as groundwater pumping (R. and P., 2005), rainwater harvesting (Li et al., 2000), changing crop use

(Kuil et al., 2018) and irrigation practices (Nouri et al., 2019b; Mollinga, 2003), are often realized at the individual or communal level. While these measures are usually beneficial for some, adverse effects can be experienced by other water users across different scales (Di Baldassarre et al., 2021). In addition, the costs and benefits of water stress-related interventions may vary throughout a heterogenous farmer population. To fully comprehend how water stress develops over time and space and to explore which interventions are suitable to reduce water stress, it is important to understand feedbacks

in a coupled human-natural system simultaneously  local household- and large basin scale.

While most hydrological models are well-suited to simulate the hydrological system at a large scale, they treat small-scale human behaviour rather simplistically and homogeneously. In these models, humans often do not learn over time and do not change their adaptive behaviour under changing water risk (Aerts et al., 2018). In reality, agents adapt to changes in their environment and also respond to each other (Wens et al., 2020). For example, a water pricing tax by the government has a

direct influence on household water use, and farmers might construct wells in response to drought events. An agent-based model (ABM) appears to be an effective tool that can be used to simulate these complex heterogeneous behaviours and feedbacks. Therefore, the research realm of "socio hydrology" has developed models that dynamically couple hydrological and agent-based models to better simulate the hydrological system as well as the behaviour of individual heterogeneous agents.

In general, two approaches can be differentiated in adding a hydrological component: using an agent-based or traditional hydrological component. In the agent-based approach, all the environmental components, such as river segments, are simulated as agents. For example, Becu et al. (2003) simulate farmers, irrigation behaviour, and crop and vegetation dynamics. Their model uses a simple routing scheme that considers water abstraction and water diversions by canal managers. Another example is Huber et al. (Huber et al., 2019), who created a basin-scale coupled model where water flows

downstream from river agent to river agent, while other agents such as farmers or water managers can abstract water from the river. In this approach, the hydrological component is usually relatively simple, largely because authors usually build the hydrological component from scratch.

The other approach is to couple an agent-based model with a more traditional hydrological model by allowing the agents to interact with its water storage. For example, the widely used MP-MAS (Schreinemachers and Berger, 2011; Arnold et al.,

2015) is coupled to WASIM-ETH, a fully distributed hydrological model. Van Oel et al. (2010) published a larger coupled



grid-based model at a 270m resolution that simulates individual farmers in a large basin using a grid size of 270x270m. This approach also benefits from ongoing methodological progress in hydrological modelling (Bierkens, 2015). Large-scale hydrological models are run at an increasingly higher resolution, while other advances, such as HydroBlocks, allow us to effectively combine grid cells into hydrological response units (HRUs) while retaining the ability to accurately simulate the
hydrological system (Chaney et al., 2016), including dynamic routing (Chaney et al., 2021).

Many agent-based models with a hydrological component were released using these methods. Some of these models simulate groups of people, such as sectors or villages (Huber et al., 2019). Other agent-based models represent single water users, such as a person or household (Schreinemachers and Berger, 2011; Wens et al., 2020; Becu et al., 2003; Arnold et al., 2015). These models are better suited to simulate individual adaptation pathways, which are often paramount in capturing
the heterogeneity of the farmer population (Wens et al., 2020; e.g., Bert et al., 2011; Tamburino et al., 2020). Yet, to simulate the effect of a single agent on the hydrological system, at least one HRU per agent is required to properly represent system feedback at the individual level (Schreinemachers and Berger, 2011). Using a gridded model, this means that the grid cell size cannot be larger than the smallest farm. This requires a large computation time and computational resources, especially in regions with small holder farms. So far, this has limited the ability of coupled models that capture the full
heterogeneity of the agent population to be effectively applied on a large scale.

We propose to resolve this issue using dynamically sized HRUs within a grid cell, with each HRU representing a single farm. Each farm-level HRU can be individually operated by an agent. This way, each individual crop field is simulated as an HRU in addition to other land use types. Due to their dynamic size (e.g., one unit for 50% of the cell and 10 units each representing 5% of the cell), CWatM can be run at a relatively coarse resolution, such as 30'' (<1km at the equator) to
simulate a large hydrological basin while allowing simulation of small and individually operated farms. Because agents can directly interact with these units (their fields), we can, for the first time, investigate the interaction between small-scale individual behaviour and large basin-wide hydrological processes.

Therefore, we present the Geographical, Environmental and Behavioural (GEB) model, named after *Geb*, the Egyptian god of the earth. The model is an ABM that is dynamically linked with a specifically adapted version of the Community Water
Model (CWatM; Burek et al., 2020). GEB can simulate large-scale hydrological processes as well as the individual behaviour of more than 10 million individual farming households and their bi-directional interactions with the hydrological system. CWatM is used as a coupled model to simulate the hydrological cycle at a grid resolution of 30'' (< 1 km at the equator). Individual farmer households (~11.1 million on a normal laptop with 10GB of free RAM) and reservoir operators are simulated as fully integrated agents that can dynamically interact with (i.e., respond to and influence) the water balance
in CWatM. Through this coupling, each individual farmer can, at a daily timestep, decide to irrigate from various sources (i.e., surface, reservoir, or groundwater). Furthermore, farmers can decide to plant and harvest crops based on the available water in their environment, the status of their crops, their risk aversion, crop price, water price, weather conditions, etc. Moreover, farmers can adapt, for example, by investing in water-saving techniques, drilling boreholes, and changing crop type. All these decisions can be made at a daily timestep.





In this work, we describe how the open-source model is set up, followed by an example application model in the heavily managed Krishna basin, which encompasses ~257.000 km$^2$ or ~8% of India's land area. Here, we simulate the adaptive patterns to water stress of ~11.1 million farmers and show how adaptation through irrigation efficiency and crop choice can influence both individual farmers as well as the hydrological system through various artificial storylines. All model code is extensively documented on https://jensdebruijn.github.io/GEB/.

## 2 Model description

The GEB model is an open-source coupled hydrological and agent-based model jointly developed at the International Institute for Applied Systems Analysis (IIASA) and the Institute for Environmental Studies (IVM, VU Amsterdam). The agent-based model can simulate millions of individual farmers in addition to other agents that interact bi-directionally with the hydrological CWatM model (Burek et al., 2020). In this manner, GEB simulates the water cycle, crop management and growth, and irrigation and reservoir management, all at a daily timestep (Figure 1). The model can be adapted to run various scenarios, both influencing the ABM (e.g., provision of subsidies to farmers or the construction of additional reservoirs) or the water cycle (e.g., varying future climate scenarios).

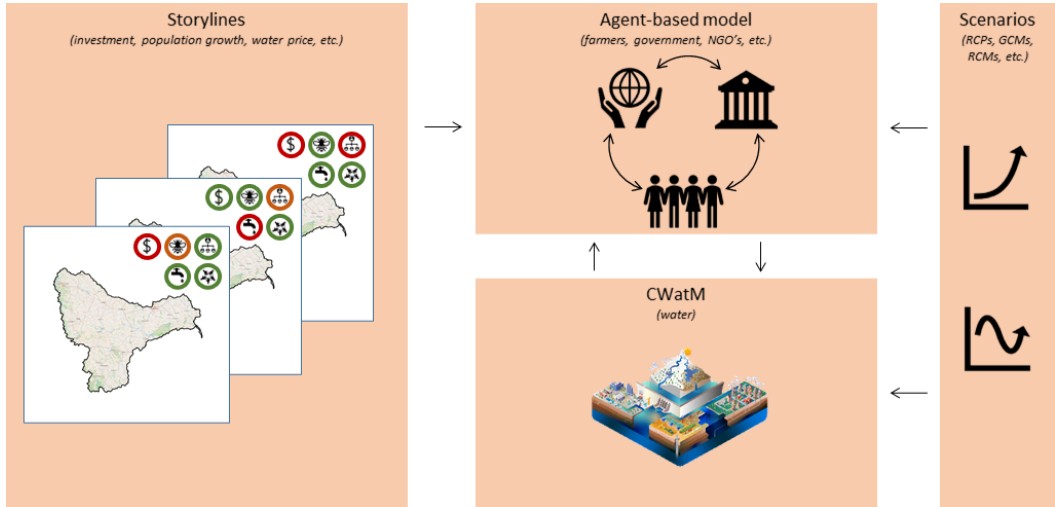

**Figure 1: GEB: High-level interaction between CWatM and the agent-based model. © OpenStreetMap contributors 2022. Distributed under the Open Data Commons Open Database License (ODbL) v1.0.**

Figure 2 shows further detail of the main interlinkages in the default setup between model components. The interactions between model components and model rules can easily be adapted for other applications or study regions. First, the model is forced by a daily set of *meteorological data*, considering the initial distribution of *land use* and *crops*. *Potential evapotranspiration* is determined for both cropland and non-cropland, which is subsequently followed by the determination of both the *water availability and potential demand*. Potential irrigation demand for non-paddy irrigation is computed as the difference between current soil moisture content and the soil moisture content at field capacity in the root zone, which is





limited by the infiltration capacity, while the potential irrigation demand for paddy-irrigated land is computed as the difference between the current water level in the paddies and the targeted water level. Here, *reservoir operators* can opt to influence water availability by releasing water from their reservoir based on considerations such as current demand and

water levels. Then, after *water consumption* by industrial, domestic, and livestock sectors, farmers can abstract irrigation water.

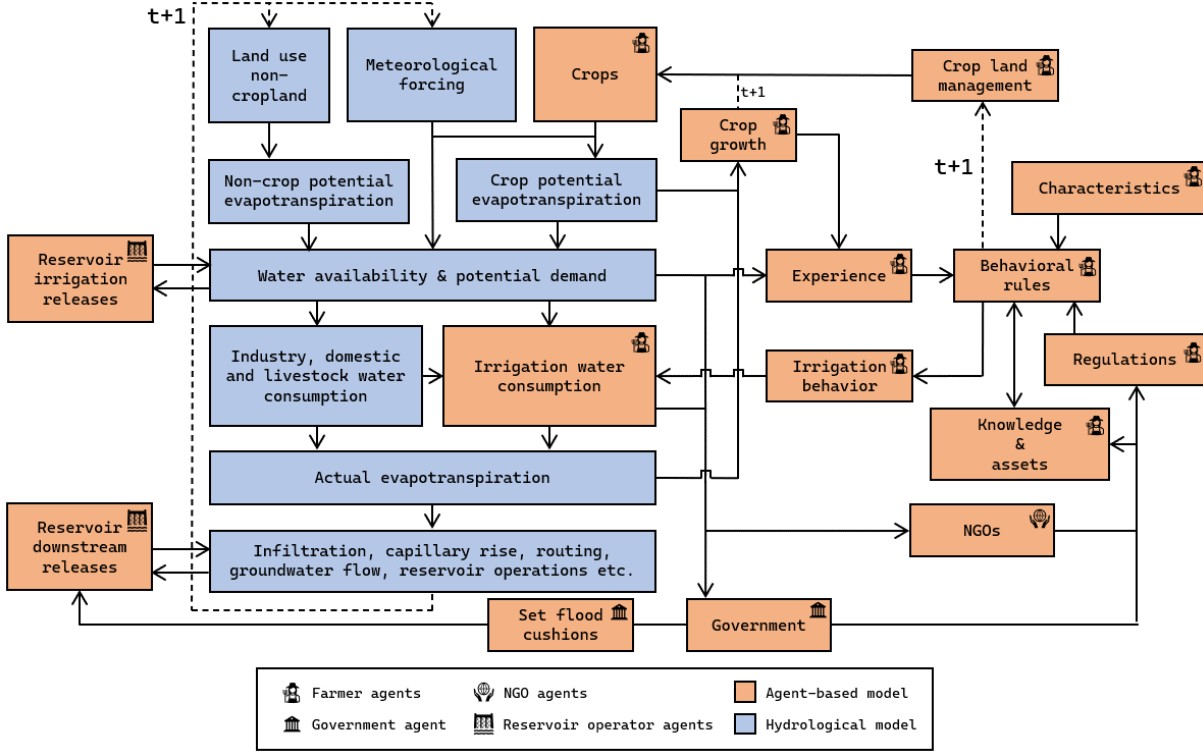

**Figure 2: A schematic overview of GEB.**

Here, the water consumption component of CWatM is adapted to interface directly with the agent-based model and thus

considers the irrigation behaviour of individual agents. This *irrigation behaviour* is, in turn, determined by crop water requirements and irrigation equipment. In future work, we will more accurately simulate farmer behaviour by including factors such as 1) current and historical experiences of water availability and water requirements, 2) agent *characteristics* (e.g., risk profile, household size), 3) agent assets (e.g., irrigation equipment, groundwater well), and 4) agent knowledge and regulations. These factors are not necessarily static over time, as agents can invest in assets (e.g., drip irrigation equipment),

farm size can change, etc. Moreover, other agents, such as government and NGO agents, can impose regulations, provide knowledge to the farmer population, or invest in the wider availability of assets (e.g., create an irrigation reservoir). Knowledge can also be obtained from other (neighbouring) agents.

After the application of irrigation water, CWatM simulates infiltration, capillary rise within soils, groundwater recharge, surface routing, and groundwater flow (through MODFLOW). Here, CWatM again communicates with the reservoir



operator agents to determine the amount of water released downstream. Then, as a new timestep is initiated, each farmer can decide to plant or harvest crops based on *experience*, *assets*, *characteristics*, *knowledge*, and *regulations*, updating the CWatM land use classes accordingly. Finally, the next timestep is initiated, starting with meteorological forcing, as described above.

The model is largely implemented in Python 3, with all computational intensive parts written in compiled Python libraries

such as NumPy (Harris et al., 2020) and Numba (Lam et al., 2015), and includes optional GPU vectorization of soil components through CuPy. The model can be run on all major platforms (i.e., Linux, Windows, and Mac). An optional model interface is extended from Mesa (Kazil Jackie and Masad, 2020; Figure 3).

**Figure 3: Optional model interface. The model can be run for one more timestep, and one can show all model variables on a map.**
**Here, the land use type is shown. The red dots represent farmer agents. Optional line charts can be added to show variables like discharge and mean groundwater level over time. For visualization purposes only, a small subbasin northwest of Pune (Maharashtra) is shown here. Land use map derived from Jun et al. 2014.**





## 2.1 Hydrological response units

Most hydrological models implement several different land use types (e.g., Burek et al., 2020; Sutanudjaja et al., 2018;
Müller Schmied et al., 2021). In these models, soil processes in all land use types are simulated individually. Runoff and
several other hydrological fluxes are then computed by aggregating to the cell level while considering the relative size of
each land use type in a particular cell. In other words, each land use type within a grid cell is simulated as a hydrological
response unit (Flügel, 1997; HRUs; Chaney et al., 2016). Farmers usually occupy cropland land use types such as non-
irrigated land, paddy-irrigated land, and non-paddy irrigated land (Burek et al., 2020; Sutanudjaja et al., 2018; Hanasaki et
al., 2018; e.g., Alcamo et al., 2003). When a single land use type within a grid cell is occupied by multiple farmers, these
farmers share an HRU (i.e., hydrological environment) and are thus simulated as a single unit (of multiple farmers).

This introduces an issue for agent-based models that focus on the implementation of heterogeneous decision-making at the
household scale. For example, when two farmers share an HRU and farmer #1 decides to irrigate while farmer #2 does not,
the soil moisture in the field of farmer #1 should increase relative to the soil moisture in the field of farmer #2. However,
when both farmers share their HRU, the soil moisture in their field cannot be separately simulated in the model.

The most straightforward solution is to run the model at a higher resolution, such that the smallest field is simulated as a
single grid cell while other larger fields are simulated as multiple grid cells. However, as small farms less than 1 ha make up
72% of global farms (Lowder et al., 2016), this solution requires the use of grid cells less than $100 \times 100$ m, which would
use an enormous amount of computational resources, making the approach unfeasible in larger basins.

As a solution, we simulate the field of each farmer as a single HRU and adapt CWatM to be able to work with these HRUs
(Figure 4). In this concept, cropland use types are further subdivided into dynamically sized HRUs based on the land
ownership (or rent) of the agent (e.g., a farmer). These HRUs can be independently operated by agents in the ABM, such as
farmers. In this manner, the land management decisions (e.g., crop planting date and irrigation) and soil processes (e.g.,
percolation, capillary rise, and evaporation) are independently simulated in an HRU for each farmer, thus allowing
simulation of multiple independently operated farms within a single grid cell. These HRUs can also be split, allowing, for
example, farmland expansion into other land use types and the sale of (part of) a farmer's land.




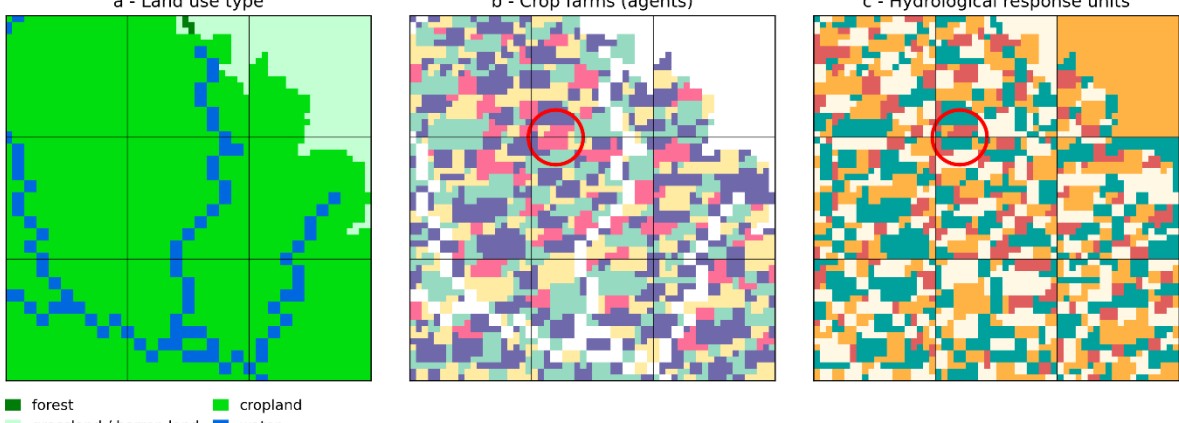

**Figure 4: In this figure, 3×3 grid cells are shown, delineated by horizontal and vertical black lines (30" resolution). Panel a displays various land use types at 20 times higher sub-grid resolution, panel b shows the crop farms owned by agents, and panel c shows the resulting HRUs. Each contiguous area of one colour in panel b represents a farm, while each contiguous area of one colour in panel c represents an HRU. One exception is non-crop HRUs of the same land use type within a grid cell, which belong to one HRU (e.g., all rivers within a grid cell are 1 HRU). Crop farms owned by one farmer that cross grid cell boundaries are represented by multiple HRUs; see, for example, the crop farm in the centre of the red circle. Land use map derived from Jun et al. 2014.**

Each crop farm that is owned by a farmer is thus an HRU. An exception is when a crop farm is spread across multiple grid cells, in which case it is represented by multiple HRUs across those grid cells. In addition, each land use type that is not operated by crop farmers in a grid cell is a separate HRU, thus, operating independently from other land use types, such as water areas, grasslands, and forests.





While most primarily "vertical" hydrological fluxes (e.g., infiltration, percolation) occur within HRUs, river discharge and
groundwater flow are simulated at the grid cell level. To this extent, conversion of fluxes between HRUs and grid cells is
required. Figure 5 shows how this works in practice, similar to hydrological models that simulate multiple land use types in a
single grid cell, such as CWatM and PCR-GLOBWB. Runoff is first determined per HRU and then aggregated to the grid
cell level while considering the relative size of each HRU. Aggregated runoff is then added to discharge, followed by solving
the kinematic wave equation at the grid cell level.

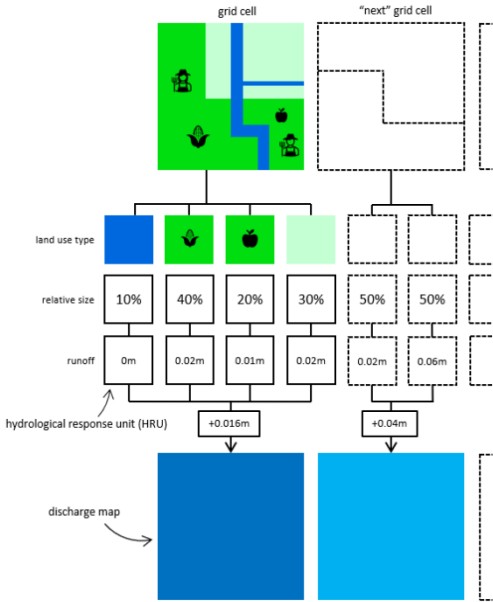


**Figure 5: Schematic overview of the implementation of farm-level HRUs. Here, a grid cell consists of four HRUs; one water-covered area, two crop farms, and one grass-covered area. Runoff is determined per HRU and then aggregated considering the relative size of the HRUs to compute runoff for the entire grid cell.**

## 2.2 Agents

The ABM currently has four types of agents: farmers, reservoir operators, a government, and an NGO. Farmers and reservoir
operators directly interact with CWatM, while government and NGO agents only communicate with other agents (e.g., by
providing subsidies to farmers). Below, we discuss the default decision-making process of agents, some of which might be
altered in storylines (see Section 0). Once behavioural rules are determined, agent behaviour is relatively easy to adapt in the
model.

Farmer agents are initialized according to a farm map, here, at a 1.5'' resolution (i.e., 20 times higher resolution than the
CWatM grid; <50 m at the equator). Given $n$ farmers, a farm map specifies the farm owned by each farmer through $n$ unique
identifiers. Correspondingly NumPy c-style arrays of length $n$ specify farmer characteristics. For example, a Boolean scalar
array with 10 million values specifies whether each of 10 million farmers has access to groundwater in the initial timestep.
Other characteristics include longitude, latitude, crop type, plant date, and harvest date.





From then forward, each farmer $F$ irrigates each HRU they occupy. Upstream agents, as determined by their elevation, are allowed to abstract water first on a "first-come, first-serve" principle. As agents have no incentive to consider environmental flow conditions, these are not enforced. Famer irrigation demand ($dem$) is determined by the difference between field capacity ($FC$) and soil moisture ($SM$) and is limited by infiltration capacity ($IC$). If farmers have access to the right equipment for surface ($F_{sw}$), reservoir ($F_{res}$), and groundwater irrigation($F_{gw}$), irrigation demand ($dem$) is then satisfied,

first from surface water (Eq. 1), then from reservoirs (Eq. 2) and groundwater (Eq. 3). All sources are limited to current water availability from the streamflow ($avail_{sw}$) in grid cell $G$, reservoirs ($avail_{res}$) that supply the command area of grid cell $G$, and groundwater in grid cell $G$. In addition, farmers only have access to water resources if they have the relevant irrigation equipment.

$$dem_{HRU} = \min (FC_{HRU} - SM_{HRU}, IC_{HRU}) \tag{1}$$

$$irr_{HRU,sw} = \begin{cases} 0, & F_{sw} = False \\ \min(dem_{HRU}, avail_{sw,G}), & F_{sw} = True \end{cases} \tag{2}$$

$$irr_{HRU,res} = \begin{cases} 0, & F_{res} = False \\ \min(dem_{HRU} - irr_{HRU,sw}, avail_{res}), & F_{res} = True \end{cases} \tag{3}$$

$$irr_{HRU,gw} = \begin{cases} 0, & F_{gw} = False \\ \min(dem_{HRU} - irr_{HRU,sw} - irr_{HRU,res}, avail_{gw,G}), & F_{gw} = True \end{cases} \tag{4}$$

When a farmer decides to irrigate, the water is subtracted from the relevant sources in CWatM and then applied to the land in the relevant HRU.

The farmer's initial crop choice and growing pattern (e.g., single or double cropping) are also loaded as an array. The planting and harvesting dates are dependent on crop type and growing pattern but, if required, could be dynamically determined by the agent, for example, on the basis of weather forecasts. Once the farmer decides to harvest their crop, the respective HRU is set to "barren land" in CWatM. Then, as the farmer decides to plant a new crop, the land use type is changed accordingly in CWatM (e.g., to "irrigated"). During a model run, farmers can decide to switch to different crops

(see Section 0).

Crop growth is differentiated into four growing periods (i.e., initial, development, mid-season, and late-season), in which crop factors are also based on the specifications of Siebert and Döll (2010). The crop yield ratio is determined based on the ratio between actual and potential evapotranspiration, also following the specifications in Siebert and Döll (2010).

The reservoir operator agents release a maximum percentage of the current reservoir water volume for irrigation purposes

each day. Hence, as upstream farmers get to abstract water first, this can lead to limited access to reservoir water for farmers at the tail end of command areas. The amount of water released for other purposes (e.g., maintaining outflow, reducing water level) depends on the rating curve of the reservoir and relevant flood cushions (Burek et al., 2020).

Finally, a government and NGO agent exist within the model. These agents can communicate with farmers and reservoir operators. In the default settings, these agents take no action but can take action depending on the model settings (see Section

240    0).



## 2.3 Model integration

This section discusses the coupling between the ABM and hydrological model, serving as both an explanation to GEB, and allowing the reader to couple their hydrological model to the ABM or vice versa. The ABM can be found in the GEB repository[1], while the adapted version of CWatM can be found in the ABCWatM repository[2]. Because both models are written in Python, the coupling is performed by *subclassing* both models and synchronizing their timesteps while adapting functions within each model to communicate with the other.

A coupling to another hydrological or agent-based model can be made by adapting the model class. It is, however, required that the hydrological model can work with farm-level HRUs. In essence, the model class 1) loads the model configuration file (see below), 2) loads a shared data *superclass*, 3) initializes the ABM, 4) initializes the hydrological model, and 5) iteratively runs a timestep of both models.

The configuration file (default: "*GEB.yml*") contains configuration parameters for both models, such as the start and end date of the simulation, which are later used in the respective models. Then, the shared data class ("`self data`") is loaded. The data class loads a mask of the study area and a land use map and automatically creates the grid and HRUs. The data class also contains convenience functions to convert data between the grid and HRUs. An example is given in Figure 6. It is useful but not necessarily required that all data for the grid cells (e.g., river discharge) and HRUs (e.g., soil moisture in the upper soil layer) used by the respective models is contained within this class, allowing easy access to these variables from both models.

---

[1] https://github.com/jensdebruijn/GEB
[2] https://github.com/jensdebruijn/ABCWatM





```
# subgrid hydrological response units (colors indicate data belonging to the same grid cell.
land_use_type = [0, 1, 3, 3, 1, 3]   # 0: forest, 1: grassland, 3: irrigated cropland
land_owner = [-1, -1, 1, 2, -1, 2]   # -1: no owner, 1: owned by farmer #1, 2: owned by farmer #2.
land_use_ratio = [.3, .7, .1, .2, .7, 1.]   # HRUs of one grid cell always sum to 1
cell_boundaries = [0, 2, 5, 6]   # 1st grid cell contains the 0th up to the 2nd HRU, the 2nd grid cell the 2nd up to the 5th, etc.

# set runoff from natural sources
runoff_m_HRU = [1, 2, 4, 1, 1, 0]

# farmer #2 has a groundwater pump running, creating additional runoff from irrigation return flow.
for i, land_owner in enumerate(land_owners):
    if land_owner == 2: # increase runoff by 1 in fields owned by farmer #2
        runoff_m_HRU[i] += 1

runoff_m_HRU
>> [1, 2, 4, 2, 1, 1]

# translate subgrid to grid – unit in meters in the example.
runoff_m_grid = []
for left, right in zip(cell_boundaries[:-1], cell_boundaries[1:]):
  runoff_m_grid.append(sum([
      runoff * land_size
      for runoff, land_size in zip(runoff_m_HRU[left:right], land_use_ratio[left:right])
  ]))

runoff_m_grid
>> [1.7, 1.5, 1]

# translate grid to subgrid – unit in meters in the example.
precipitation_m_grid = [1, 3, 2]
precipitation_m_HRU = []
for left, right, precipitation_HRU_m in zip(cell_boundaries[:-1], cell_boundaries[1:], precipitation_m_grid):
  number_of_HRUs_in_grid_cell = right – left
  for i in range(number_of_HRUs_in_grid_cell):
    precipitation_m_HRU.append(precipitation_HRU_m)

precipitation_m_HRU
>> [1, 1, 3, 3, 3, 2]
```

**Figure 6: A simplified code example of the conversion between hydrological response units (HRUs) and the grid.**

Then, the agent-based model is initialized with a set of agent attributes. For farmers, this usually consists of a raster map that indicates the area that is managed by a specific farmer, the locations of the farmer (e.g., the centre point of the field) and other attributes, such as crop type, cropping schedule and irrigation status. Finally, CWatM is initialized as per Burek et al. (2020) while loading initial land use and crop parameters from the ABM.

# 3    Model application

Here, we show the application of the model in the heavily managed Krishna basin in India. The main aim here is not to perform a fully realistic analysis of agent behaviour but rather to showcase the model by showing its ability to simulate more than 10 million farmers and to show how various artificial storylines influence model results for agents as well as river discharge.

## 3.1 The Krishna basin

With a size of roughly 8% of India's land area, the Krishna basin in India (Figure 7) is a complex socio-ecological system experiencing several sustainability and equity challenges, particularly related to water management. The basin is important for agricultural production while being exposed to floods, droughts, and dropping groundwater tables (Surinaidu et al., 2013). A large number of reservoirs with a total volume of approximately 42 billion m$^3$ (~20% of annual rainfall) were built



primarily for irrigation purposes. Farmers in a reservoir command area can access the reservoir water distributed through a system of canals. In addition, following the Indian Agriculture Census[3], approximately 8% of farmers have access to groundwater through a well, depending on the farm size and location. The mean farm size is ~1.5 ha.

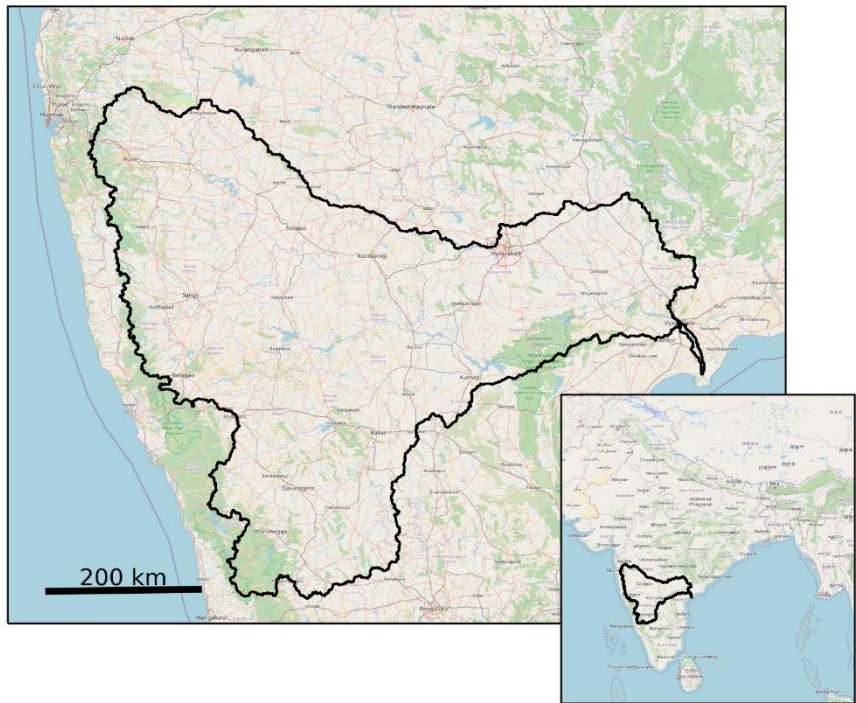

**Figure 7: Outline of the Krishna Basin in India. © OpenStreetMap contributors 2022. Distributed under the Open Data Commons Open Database License (ODbL) v1.0.**

**3.2 Model setup**

First, we selected the region basin for the study using the MERIT Hydro elevation map (Yamazaki et al., 2019), which was upscaled to 30" using the Iterative Hydrography Upscaling method (Eilander et al., 2020a), and subsequently selecting all upstream cells of the Krishna river outlet. Other routing maps, such as river slope and width, were obtained similarly (Eilander et al., 2020b). Reservoir and lake footprints were obtained from the HydroLAKES dataset (Messager et al., 2016).

If available, flood cushions and reservoir volumes were obtained from the Andhra Pradesh WRIMS[4]. If not available, flood cushions were assumed to be zero, while reservoir volumes were taken from the original HydroLAKES data. Reservoir command areas were obtained from the India Water Resources Information System (India-WRIS) and subsequently manually linked to the previously obtained reservoir using satellite imagery. Reservoir operator agents are assumed to release a maximum fraction of the current reservoir volume for irrigation, limited by the irrigation demand in the command

---

[3] http://agcensus.dacnet.nic.in/
[4] https://apwrims.ap.gov.in/





area. Land use was obtained at 30-meter resolution from GlobeLand30 (Jun et al., 2014), downscaled to 1.5" and mapped to CWatM land use types. Pixels that were classified as "water body" in GlobeLand30 and all cells with at least 100 km² upstream area were classified as "water covered area" in CWatM. All other input data were obtained from CWatM input maps at 5' resolution and downscaled to 30" for CWatM input. The groundwater MODFLOW model is defined by an orthogonal grid at a 1000m resolution. Only one homogeneous unconfined aquifer layer is considered. One pumping well is

set up in each MODFLOW cell to satisfy the water demand from farmers and other sectors.

Water demand and consumption for industrial, domestic, and livestock sectors are estimated using the approach developed by Wada et al. (2011) and are then downscaled to the size of the land units by distributing the demands over cells with relevant land uses; grassland for livestock demands, and sealed area for industrial and domestic demands. The model was forced with GSWP3 (Dirmeyer et al., 2006) provided within ISIMIP3a.

**3.3  Creating farmer agents**

As a map of individual farms is not available to the best of our knowledge, we created a synthetic map of individual farms and corresponding agents in the area designated as cropland in GlobeLand30. The aim here was not to create a fully accurate farm map but one that is statistically representative of the area. To do so, we obtained farm sizes at the district level from the Indian Agricultural Census[5]. Then, we randomly generated farms (see example Figure 4b) while considering the distribution

of farm sizes, resulting in approximately 11 million individual farms and corresponding farmer agents. We generated individual farmer access to surface water irrigation using a map of irrigated areas in 2010-2011 (Ambika et al., 2016). In addition, some irrigating farmers had an irrigation well assigned using probabilities of having a well for each farm size class per the agricultural census at the district level[5].

Farmers are then assigned a crop and planting and harvesting scheme (single, double, triple-cropping) based on their

irrigation status and location using the MIRCA2000 dataset[6] (Siebert and Döll, 2010). Here, the MIRCA2000 crop areas are downscaled to the resolution of the farmer map, and the crop and crop planting scheme are then randomly sampled based on the location of the farmer and the relative area of all crops within that grid cell.

**3.4  Calibration**

The model is calibrated based on daily river discharge from the India Water Resources Information System (WRIS) for the

Wadenepally station in the Krishna River, roughly 60 km upstream of Vijayawada. Calibration is performed based on several hydrological parameters (Burek et al., 2020), as well as the maximum amount of water released from a reservoir for irrigation purposes on a given day, the normal reservoir outflow, and the irrigation return fraction, using the NSGA-II genetic algorithm (Deb et al., 2002) as implemented in DEAP (Fortin et al., 2012). The modified version of the Kling-Gupta

---

[5] http://agcensus.dacnet.nic.in/
[6] https://www.uni-frankfurt.de/45218031/data_download




efficiency score (KGE; eq. 5; Kling et al., 2012) is used as an objective function with $r$ as the correlation coefficient between

monthly simulated and observed discharge, $\beta = \frac{\mu_s}{\mu_o}$ as the bias ratio, and $\gamma = \frac{CV_s}{CV_o} = \frac{\sigma_s/\mu_s}{\sigma_o/\mu_o}$ as the variability ratio.

$$KGE' = 1 - \sqrt{(r-1)^2 + (\beta-1)^2 + (\gamma-1)^2} \tag{5}$$

where $r$, β, and γ are all optimal at 1. The period 2004-2006 is used as a spin up period, 2006-2012 is used for calibration, and 2012-2018 is used as the test period.

The genetic calibration algorithm first generates 60 parameter sets within a predefined range of plausible options (i.e., "the

population"), and the model is subsequently run for each parameter set (i.e., "individual"). Then, the 10 most optimal parameter sets are combined (i.e., "mated") with a probability of 0.7 or altered (i.e., "mutated") with a probability of 0.3 to create 12 new parameter sets for which the model is also run. Then, the 10 most optimal parameter sets are selected again from all previous model runs, including the initial set. After 10 iterations (i.e., "generations"), the most optimal parameter set is finally selected.

**3.5 Storylines**

After calibration, we test how the model responds to various endogenous and exogenous drivers of adaptation, running $3 \times 2$ simple storylines with a combination of exogenous and endogenous drivers of adaptation. Note again that these storylines are not meant to be actual projections of reality but rather to showcase the model.

Exogenous drivers:

1. **No adaptation**: all farmers have an irrigation efficiency of 60%, meaning that 60% of irrigation water infiltrates the soil while 40% is return flow.
   2. **NGO adaptation**: an NGO is assumed to disseminate knowledge about how to improve irrigation efficiency to 80% for 100.000 farmers. All farmers with a higher irrigation efficiency have a daily 1% probability of disseminating the knowledge to another farmer within a 5 km range (i.e., to simulate a social network).
3. **Government subsidies**: the government provides subsidies to improve irrigation efficiency to 80% for 5% of farmers each year.

Endogenous drivers:

   1. **No crop switching**: farmers stick to the crops that they initially use.
   2. **Crop switching:** each year, farmers with a field size of at least 0.5 hectares have a 30% probability of switching to
sugar cane, a high water consumptive crop, if they were water limited for less than half of the days during the previous growing season.

For all storylines, we use a spin up period of 2012-2014 and show model output for 2014-2018.

**4  Results and discussion**

Figure 8 shows observed versus simulated discharge in $m^3/s$ for the calibration model. The KGE during the calibration

period is 0.849, while the KGE during the test period is 0.709 (1 is optimal), showing a good calibration performance for the





model. Figure 9 shows irrigation from channels, reservoirs, and groundwater for all agents. While the data is difficult to show for individual farmers at this scale, the insets show the detailed heterogeneous irrigation quantities at the field scale for a small portion of the basin. Yet, on the larger scale, it is clearly visible that farmers along rivers and within reservoir command areas have better access to irrigation.

Yet, we should also note that agent behaviour is rather simplistic and homogeneous in this conceptual model, and should be made more realistic using empirical parameterization in future research. This requires data campaigns and stakeholder engagement, specifically looking into the current state of farmer crops, crop management and the irrigation options and behaviour of agents. Furthermore, adaptation options (e.g., crop switching, well digging, drip irrigation, rainwater harvesting) need to be properly considered (Tamburino et al., 2020), by including factors such as threat appraisal (e.g.,

perception of drought risk), the coping appraisal of individual farmers (e.g., knowledge, information, and financial resources; Wens et al., 2020; Schrieks et al., 2021), farmer networks (Wens et al., 2020) and collectives (Shah and Bhattacharya, 1993). In addition, the automatic delineation of small-holder fields using machine learning (Waldner and Diakogiannis, 2020) and the recognition of crop types at the field scale (Gumma et al., 2020) could benefit a more realistic model.

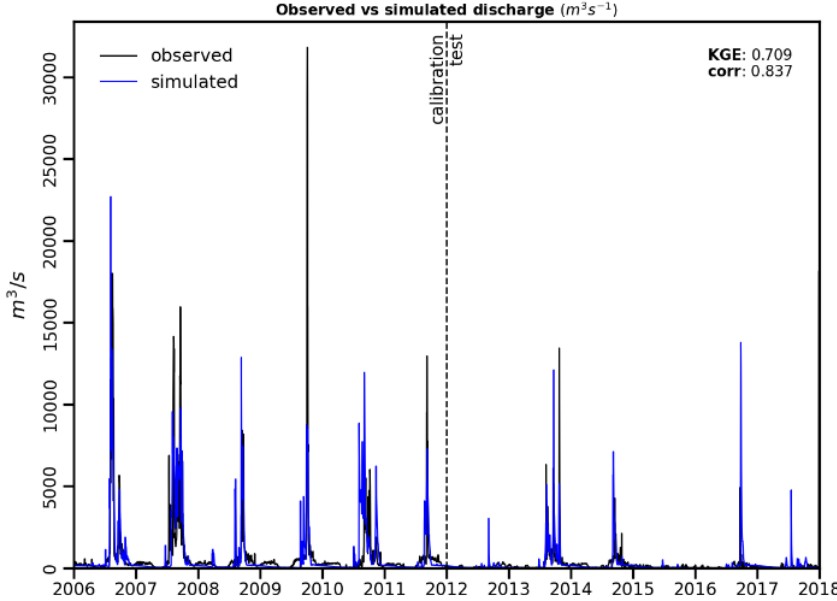

**Figure 8: Observed versus simulated discharge for the calibrated model.**

After further development of agent behaviour, new calibration and validation targets would also be opened up, such as the availability of soil moisture observations from satellites at very high resolution. Furthermore, farmer characteristics over time can be calibrated based on tehsil-level (local administrative unit in India) census data on irrigation type, crop type etc.



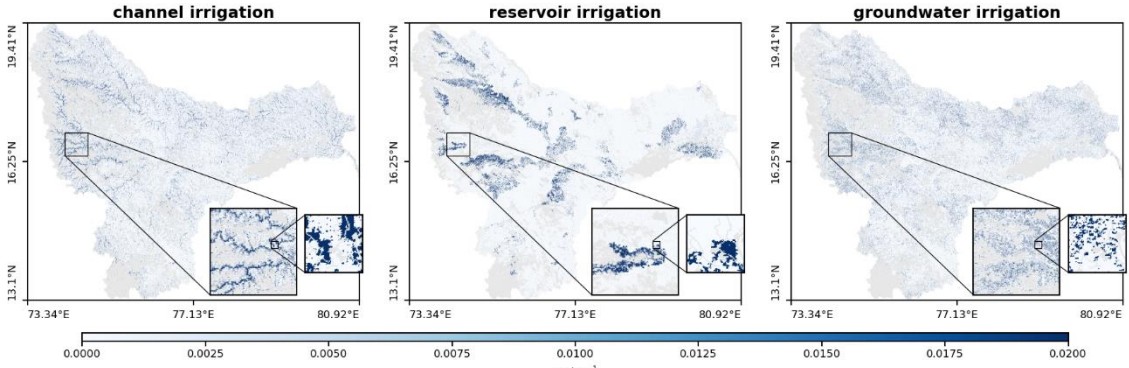

**Figure 9: Average daily irrigation from channels, reservoirs, and groundwater.**

## 4.1  Computational requirements

The model for the entire Krishna basin can be run on an above-average laptop. Model run time was ~10 seconds per daily timestep (i.e., ~1 hour for one year) using a single core on an AMD EPYC 7302 while requiring no more than 7GB of RAM and an 8GB RTX1070 GPU. Without GPU, the run time was ~30 seconds per timestep while requiring 12GB of RAM.

Model run time and requirements scale near-linearly with basin size, assuming identical farm sizes. Larger farm sizes reduce the requirements, while smaller farms increase the requirements. Here, the average farm size is 1.6 ha.

## 4.2  Storylines

Finally, Figure 10 shows the potential of the model to simulate the storylines (see Section 0). Panels D and E show the percentage of farmers that have adopted both high irrigation efficiency and sugar cane crops. In the storylines with no

irrigation adaptation, all farmers remain at low irrigation efficiency, while an increase is observed in the other storylines. For the adoption of sugar cane, it can be seen that a slightly higher percentage of farmers adopt sugar cane in the storyline with NGO adaptation (blue line) compared to the storyline with government subsidies (orange line) because adoption of high irrigation efficiency is faster in the NGO adaptation storyline. This increases the availability of water, and hence more farmers switch to sugar cane.

The effect of these storylines is also clearly visible in panel C, where the mean hydraulic head across the entire region is shown. A lower hydraulic head indicates a lower groundwater table. Clearly, the most sustainable storyline—with the least dropping groundwater table—is the one where farmers do not switch crops but do increase their irrigation efficiency. In contrast, the storyline with both low irrigation efficiency and crop switching is the least sustainable. Similar effects, albeit less clear, can be observed in the discharge (panel A) and reservoir storage (panel C). Especially at the end of the dry season

(see insets in panel C), reservoir storage is lower for the less sustainable storylines and vice versa. Similarly, the reservoirs fill up quicker in the sustainable storylines. While discharge is difficult to distinguish between the storylines, it can also be seen in the insets of panel A that the discharge is slightly higher at the end of the dry seasons. Similarly, the discharge peaks





are higher during the wet season in the more sustainable storylines. While any increased flood risk from these discharge peaks can easily be averted by increasing the size of the flood cushions or increasing the release of water downstream when

the reservoir is near-full, this could lead to a slightly higher flood risk if identical reservoir operation rules are used in all storylines.





**Figure 10:** This figure shows several model outputs for six storylines. Panel A shows daily discharge, panel B shows the mean hydraulic head, panel C shows total reservoir storage, panel D shows the percentage of farmers with a high irrigation efficiency, and panel E shows the percentage of farmers that adopted sugar cane. Note that the y-axes for panels D and E are different.



## 5       Conclusions

Here, we present for the first time a coupled agent-based and hydrological model that can simulate millions of individual households and their bi-directional interactions with the hydrological system while allowing for the assessment of large-scale hydrological processes. By adopting a fully distributed hydrological model with dynamically sized farm-level hydrological

response units (HRUs), the farmers can individually interact with the hydrological model while keeping computational demands reasonable. This opens up opportunities to study large- and small-scale socio-hydrological processes simultaneously.

Future research is still required to further simulate the behaviour of farmers on a large scale. Future studies can also further investigate how governments or other organizations can affect the human-natural system, such as through reservoir

construction and management, water/electricity pricing (Parween et al., 2021), water rights (Nouri et al., 2019a), enforcing specific crop types (Wallach, 1984), etc.

Further coupling to a hydrodynamic model such as DIM (Farrag et al., 2021) allows researchers to investigate the interactions between human behaviour and flood and drought risk (Ward et al., 2020). They can also enhance agent behaviour by coupling this behaviour with economic models or models that simulate land use change (Dou et al., 2020). In

addition, the integration of future scenarios such as climate change, population growth, and exogenous land use can be used to project how the coupled human-natural system is projected to change into the future.

## 6    Code availability

All     model     code     is     available     from     https://github.com/jensdebruijn/GEB     for     the     coupled     model (https://doi.org/10.5281/zenodo.6817560), https://github.com/jensdebruijn/ABCWatM for the adapted version of CWatM

(https://doi.org/10.5281/zenodo.6817570)  and  https://github.com/jensdebruijn/honeybees  for  the  agent-based  modelling environment (https://zenodo.org/record/6817568).

## 7    Data availability

All input data for GEB can be obtained from the original data source as described in the documentation. Scripts for downloading and processing data are provided in the 'preprocessing' folder. Data for CWatM is similarly described in the

documentation, and can also be obtained from the IIASA FTP server.

## 8    Author contribution

JB designed and coded the main model, MS designed and coded the initial hydrological model used in the Krishna basin, PB is the main author CWatM and provided input for adapting CWatM to its agent-based version, LG contributed the




groundwater module, YW and JA contributed to conceptualization and methodology. JB prepared the manuscript with
contributions from all co-authors.

## 9   Competing interests

The authors declare that they have no conflict of interest.

## 10   Acknowledgements

This research received funding from  IIASA's Strategic Initiatives Program through the fairSTREAM project and from the
European Research Council through the ERC Advanced Grant project COASTMOVE (grant number 884442).

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
