# Peer review of "GEB v0.1: A large-scale agent-based socio-hydrological model – simulating 10 million individual farming households in a fully distributed hydrological model"

_EGUsphere, 2022_

## Author Response (AR1)

**Reviewer #1**

*We thank the reviewer for taking the time to review our manuscript. Below we list the comments made by the reviewer and suggest changes in italic.*

The study by Bruijn et al. with the title of "A large-scale agent-based socio-hydrological model – simulating 10 million individual farming households in a fully distributed hydrological model" is intended to provide a coupled agent-based and hydrological model to simulate farmers' behavior. At the current stage, the paper does not present a proper understanding of agent-based modeling and socio-hydrological model. While the study implies that agent-based modeling benefits this work, I am quite concerned and surprised about the materials of the paper regarding agent-based modeling:

- What is the difference between your current model and the high-resolution water management model? The authors named some components as agents (i.e., reservoir operators, a government, and an NGO) though they act as the same as a traditional water management model. For example, there is a so-called NGO agent, but when we read the information about its action in Section 3.5, there are like "scenarios"! the same example can be seen for the government agent. Thus, another concern can arise regarding the wrong definition of agents in this study.

*CWatM does not include the high-resolution water management components which are described in this manuscript i.e., CWatM only has grid-cells, and no sub-grid information / hydrological response units. The differences between CWatM and the high-resolution water management model are described in section 2.1 and 2.2. These sub-grid hydrological response units allow the simulation of these individual farmers agents (of which there are >10 million) and their individual bi-directional connection to the fully distributed hydrological models. The NGO agent, government agent and reservoir operators or entities can make individual autonomous decisions that affect farmer agents or the environment (and thus other agents indirectly), and are thus agents* (Bonabeau, 2002).

*However, it is indeed correct that the behaviour of farmer agents is rather homogeneous. Our intend was to present the model framework first which includes a framework for computationally optimized ABMs in Python, a heavily adapted hydrological model (e.g., sub-grid hydrological response units), and introduce several stylistic scenarios to showcase the model. However, based on the reviewer's comments (as well as other reviewers) we propose to replace the current scenarios (with rather homogeneous agent characteristics) with a scenario with more heterogeneous agent characteristics (and resulting behaviour). We hope that this implementation will make the agent-based modelling component stronger, and will thus satisfy the reviewer. To do so, we will*

1. *Use the Indian Development Human Development Survey (IHDS), which presents highly detailed information on 41.554 households, including crop types for the different growing seasons, household size, household income, expenditure, irrigation techniques, farm size etc etc.*
2. *From the Indian agricultural survey, we will collect farm characteristics (i.e., marginal distributions), including farm sizes and crop types at tehsil level (comparative to counties in the US)*
3. *Use an adapted version of the iterative proportional fitting to create a synthetic farmer population using the micro-level data from the IHDS yet fits the marginal distributions of the tehsil level agricultural survey*

Then based on this data, land use data derived from satellite imagery, and the distribution of farm sizes, we will distribute those heterogeneous farmers spatially, which now include heterogeneous characteristics (according to the IHDS), such as crop types for the different growing seasons, household size, household income, expenditure, irrigation techniques, farm size etc.

Then, using historic crop prices, as obtained from the Indian Agricultural Marketing Information System, in combination with simulated yield based on farm size, potential yield and the simulated ratio of actual to potential evapotranspiration, we will simulate individual farmer income. Combined with inflation-adjusted household expenditures (from IHDS data) and crop expenditures (based on data obtained from the Ministry of Agriculture and Farmers Wellfare) we can calculate disposable income.

Next, we can include farmer adaptation behaviour, specifically the construction of irrigation wells, in the model. To simulate this, farmers without an irrigation well, look to (farm-size adjusted) income of neighbouring farmers with similar crop types but with an irrigation well. If the income difference of the agent's farm compared to those surrounding farms is higher than the implementation and upkeep costs of an irrigation well, the farmer will implement an irrigation well.

This also allows us to derive more in-depth conclusions, based on the heterogeneity of these agents (in addition to the presentation of the model framework and potential future applications).

- The agent-based model is well-known for stochastic processes, learning and adaptive procedure, and complex interactions among agents. How do you benefit from each or some of these features in your model structure and equations? Please explain each feature, if there is any, according to your equations.

*Stochasticity*

In short, the placement of agents and assigning characteristics of those agents are random, while decision rules are largely deterministic. Models that use agents, are on a range between fully stochastic agent-based models where all decisions are stochastic to more deterministic agent-based models, which is frequently the case for spatial models. There are many different definitions of what constitutes an agent-based model. Here, we take the definition that an agent-based model consists of many interacting entities (here: >10 million farmers), in a social setting with relatively simple rules, yet creating a complex system. Because the model is so large, the outcome of the system is unknown beforehand, and the model can provide explanatory insight into emergent behaviour of the system, similar to other agent-based models and environmental models such as traditional hydrological models and climate models. The model now includes a social component through including individually endogenously adapting agents.

*Learning and adaptative procedure*

The agents (both in the 'old' scenarios in the current manuscript and the 'new' proposed scenario as suggested above), agents can adapt over time by implementing adaptation measures such as more efficient irrigation techniques (old scenario) and the construction of groundwater wells (new scenario).

Both in the 'old' and 'new' scenarios there are several behaviours that depend on the neighbours, such as the learning of adaptation techniques and the viewing of income of neighbouring farmers. But most importantly, the behaviour of agents affects other agents through the hydrological modelling. If an upstream farmer abstracts water from a reservoir, or stream it cannot be abstracted

*by downstream farmers. Moreover, groundwater extraction, lowers the groundwater table, making investing in groundwater wells more expensive.*

***Complex interactions***

*The complex interactions between agents in this model can be subdivided in the parts:*

1. ***Agent-agent interactions:*** *In the 'old' scenario, the agents learn from each other in the NGO adaptation scenario "All farmers with a higher irrigation efficiency have a daily 1% probability of disseminating the knowledge to another farmer within a 5 km range", and in the "new" scenario, a similar procedure is proposed for adaptation of irrigation wells (see above).*
2. ***Agent-environment-agent interactions:*** *The most complex interactions between agents are through the hydrological model (e.g., when one farmer abstracts water, the "same" water cannot be abstracted by another farmer). These interactions go through the groundwater (i.e., MODFLOW), sub-surface and surface (i.e., CWatM). All these models are dynamically coupled thus creating a complex system.*

Besides these comments, I also have other major comments:

- The paper lacks a literature review on socio-hydrological models.

  *We will include an additional paragraph mentioning the benefits that coupled agent-based models have with respect to agent-based models or hydrological models only. However, we would not propose to include a full literature review in line, which is in line with other manuscripts that I have been published in Geoscientific Model Development (but instead focus on presenting the model).*

- The socio-hydrology is mainly about the coevolutionary behavior between the hydrological and human systems. Please clarify the bidirectional feedback between a hydrological and social system in a figure. The current figures do not satisfy this need. Please also note that scenarios cannot show the "co-evolutions" between the systems.

  *In an updated manuscript we will include an additional figure to show the irrigation water consumption and the effect on hydrology in more detail, see image below. Other feedbacks between the hydrological and human system, such as crop choices, and reservoir management are shown in Figure 2.*

[Figure]

- Another point of socio-hydrology is to involve social factors. In lines 131-134, I am very surprised that the authors just mentioned they will work on it in their future work. Then, what are the social components in this study?

  *These social components were included to a limited extend in the NGO scenario (lines 337-339), but will be included more in the new storyline as suggested above. Farmers will look at the income of neighbouring farmers with similar crops, and decide based on that, whether to adopt an irrigation well (more info see above).*

- I wonder if Figure 2 shows the complete picture of the current paper?! The authors mentioned some components (e.g., characteristics and experience) will be in their future work (lines 131-134)! Also, there should be a meaningful connection between Figure 2 and the equations. You should completely clarify each component of the figure and refer to the corresponding equation.

  *Here, we propose the same Figure as above, with the equations labelled as EQ1 … EQ4.*

[Figure]

- It seems that Equations 1-4 represent all agents' equations in the model. Once again, I wonder how this work benefits from the concept of an agent-based approach. what is the difference between this model and a traditional water management? It seems that the only advantage of this water management work is to provide a high-resolution model.

  *There are several differences. 1) the ability to include heterogeneous decision making, which would be realized in the newly proposed scenario and 2) the two-way interaction between the hydrological model and the social system (although this can also be included in other types of models, such as system dynamics models) this is a strength of the integration between ABMs and hydrological models).*

- I am very surprised by the conclusion section. Around 75% of the conclusion is about future work! what is the take-home message of this study? What are the implications?

  *The main aim of the manuscript is to present the model, which for the first time enables highly detailed simulation of millions of farmer agents in large river basins. This model can be adapted for a wide range of applications, which is why we suggested several avenues of research. However, in a potential new version of the manuscript we will expand the conclusion to include several take home messages based on the newly proposed scenario, for example focussing on the heterogeneity in farmer characteristics and the influence on adaptation behaviour.*

Some minor comments:

Line 55: very confusing. Are you talking about approaches of modeling human systems? what is the traditional hydrological component as an approach? here I suggest you write about approaches of simulating water-human systems (including SD and AB).

*We are writing about the hydrological component, which is mentioned at the beginning of sentence. However, we suggest to write this more clearly as follows:*

> *"In general, two approaches can be differentiated in adding a hydrological component: using an agent-based or traditional hydrological component."*

> →
[Figure]

> *"In general, two approaches can be differentiated in adding a hydrological component to coupled agent-based hydrological model: using a hydrological component which is agent-based (e.g., river segments are represented as agents which exchange water) or traditional hydrological component (e.g., a gridded model where water flows from once grid cell to another based on the kinematic wave equation)."*

Line 63: The other approach to consider/model what?

*The previous paragraph mentions there are two approaches, we first mention the first approach, then the other approach is the second approach. To make this more clear, we suggest the following (additions in red). This also includes merging the paragraphs, making it more clear that these approaches are connected.*

*In the first approach, the agent-based approach, all the environmental components, such as river segments, are simulated as agents. For example, Becu et al. (2003) simulate farmers, irrigation*

*behaviour, and crop and vegetation dynamics. Their model uses a simple routing scheme that considers water abstraction and water diversions by canal managers. Another example is Huber et al. (Huber et al., 2019), who created a basin-scale coupled model where water flows downstream from river agent to river agent, while other agents such as farmers or water managers can abstract water from the river. In this approach, the hydrological component is usually relatively simple, largely because authors usually build the hydrological component from scratch. The  second approach, the hydrological model approach, is to couple an agent-based model with a more traditional hydrological model by allowing the agents to interact with its water storage.*

Lines 64-65: Is it an example of coupling the agent-based modeling and a hydrological model? what is the agent-based modeling part about? Please explain more.

*The aim of these paragraphs is to talk about the coupling of the hydrological components (which is also the main aim of this manuscript). Therefore, we feel it will confuse the reader to go in depth on the agent-based component of these papers. However, references are provided.*

Line 66: What does it exactly simulate?

*The irrigation behaviour. We will add this as follows (additions in red):*

*"grid-based model at a 270m resolution that simulates the irrigation behaviour of individual farmers in a large basin"*

Lines 63-70: this paragraph just provides a list of references and confuses readers. what is the main idea of this paragraph? if you want to mention the advantage of coupling agent-based modeling and hydrological model, explain the agent-based modeling and hydrological part in each study. what was the benefit of agent-based modeling for each study?

*The aim of this paragraph is to talk about the coupling mechanisms of the models, which should be clearly with the additions described above. In the updated manuscript, we will, however, include an additional paragraph which mentions some of the benefits and processes that agent-based models have included.*

Line 71: what are the "these methods"?

*This is indeed phrased in a confusing way, and suggest the following change:*

*"Many agent-based models with a hydrological component were released using these methods."*

→

*"A large number of agent-based models with a hydrological component were released."*

Lines 230 and 240: what is section 0?

*There was an error in the internal referencing in the document, this should point to section 3.5, and will be corrected.*

Line 330: this section should change to scenario analysis

*We will adapt the use of the word storyline to scenario in the entire manuscript.*

Lines 331-347: what is the argument behind choosing these numbers like 30% probability of switching crops? Do you have any references for them? If not, there is a need to do a sensitivity analysis.

*In the new scenario these will be replaced.*

Lines 401-404: "for the first time"?

*We did not try claim to present the first model that makes a coupling to a hydrological model (also see cited models in the introduction). However, to the best of our knowledge this is the first model to do so which can include "millions of individual households" in combination with "bi-directional interactions with the hydrological system". Therefore, to avoid confusion we have revised this as follows:*

*"Here, we present a coupled agent-based and hydrological model which for the first time allows simulation of millions of individual households and their bi-directional interactions with the hydrological system while assessing large-scale hydrological processes."*

**Reviewer 2**

In this paper, a large-scale agent-based model of socio-hydrological dynamics is introduced. The proposed modelling tool is scientifically promising, and the manuscript is technically sound. Yet, I see a number of shortcomings. Thus, I provide some comments that I think should be addressed before publication.

*Dear reviewer, we thank you for the kind words on the scientific promise and technical soundness. Below we suggest improvements to the manuscript based on the reviewer's comments in italic.*

1. The need for large-scale agent-based modelling is not well argued for in the introduction. This also applies to model assumptions. The paper lacks logical justifications for temporal and spatial scales as well as about the focus on the selected agents.

   *Dear reviewer, we thank you for the suggestion and agree that this justification was not clearly included. Therefore, we suggest to include the following paragraph in the introduction.*

   *Moreover, the hydrological system includes many connections across large scales, and consider farmer heterogeneity. For example, farmers at the head-end of a command area have access to a much larger and reliable water supply than tail enders* (Mollinga, 2003)*. These incentivizes head-end farmers to adopt water-intensive high-return crops, reducing water availability downstream* (Wallach, 1984)*. Similarly, upstream famers that invest in rainwater harvesting techniques reduce the amount of water available downstream* (Bouma et al., 2011)*. Yet another example is through groundwater use, where individual well users lower the groundwater table in the entire region* (R. & P., 2005)*. And while some farmers might be able to invest in deeper wells, other farmers are left behind driving them further into poverty* (Batchelor et al., 2003)*.*

In the introduction, the authors state that (line 46): "while most hydrological models are well-suited to simulate the hydrological system at a large scale, they treat small-scale human behaviour rather simplistically and homogeneously…" I cannot agree more. Yet, I was disappointed when I read that only "in future work, we will more accurately simulate farmer behaviour by including factors such as… These factors are not necessarily static over time" (definitely not!) ", as agents can invest in assets (e.g., drip irrigation equipment), farm size can change, etc. Moreover, other agents, such as government and NGO agents, can impose regulations, provide knowledge to the farmer population, or invest in the wider availability of assets (e.g., create an irrigation reservoir). Knowledge can also be obtained from other (neighbouring) agents." Well, I expected to see (at least some of) these aspects/feedbacks included into this model. Why not? Why only for future work? Without them, the modelling exercise essentially become a downscaling exercise. Is this enough to justify novelty? I am not sure.

*Dear reviewer, we thank you for your comment (which is also in line with the suggestions of other reviewers), and therefore, we suggest to include a new scenario which considers farmer heterogeneity as follows:*

4. *Use the Indian Development Human Development Survey (IHDS), which presents highly detailed information on 41.554 households, including crop types for the different growing seasons, household size, household income, expenditure, irrigation techniques, farm size etc etc.*

5. *From the Indian agricultural survey, we will collect farm characteristics (i.e., marginal distributions), including farm sizes and crop types at tehsil level (comparative to counties in the US)*

6. *Use an adapted version of the iterative proportional fitting to create a synthetic farmer population using the micro-level data from the IHDS yet fits the marginal distributions of the tehsil level agricultural survey*

*Then based on this data, land use data derived from satellite imagery, and the distribution of farm sizes, we will distribute those heterogeneous farmers spatially, which now include heterogeneous characteristics (according to the IHDS), such as crop types for the different growing seasons, household size, household income, expenditure, irrigation techniques, farm size etc.*

*Then, using historic crop prices, as obtained from the Indian Agricultural Marketing Information System, in combination with simulated yield based on farm size, potential yield and the simulated ratio of actual to potential evapotranspiration, we will simulate individual farmer income. Combined with inflation-adjusted household expenditures (from IHDS data) and crop expenditures (based on data obtained from the Ministry of Agriculture and Farmers Wellfare) we can calculate disposable income.*

*Next, we can include farmer adaptation behaviour, specifically the construction of irrigation wells, in the model. To simulate this, farmers without an irrigation well, look to (farm-size adjusted) income of neighbouring farmers with similar crop types but with an irrigation well. If the income difference of the agent's farm compared to those surrounding farms is higher than the implementation and upkeep costs of an irrigation well, the farmer will implement an irrigation well.*

*This also allows us to derive more in-depth conclusions, based on the heterogeneity of these agents (in addition to the presentation of the model framework and potential future applications).*

2. While the storylines provide a range of interesting scenarios, I understand that the key drivers are primarily exogenous (rather than endogenous). This is a missed opportunity as emerging behaviour, patterns, and surprises are the essence of sociohydrology and agent-based modelling. Can the authors clarify that?

*Thank you for your comment, which we hope is addressed is by the scenario described above.*

3. In the concluding part of the paper, it would be appropriate to discuss the results in view of the scientific community. What is the novel contribution to sociohydrology and agent-based modelling? What are the implications of this work, and what shall be done differently in future studies?

*Thank you for your comment, and we will discuss these points in more detail, also based on the newly presented scenario.*

Batchelor, C. H., Rama Mohan Rao, M. S., & Manohar Rao, S. (2003). Watershed development: A solution to water shortages in semi-arid India or part of the problem? *Land Use and Water Resources Research*, *3*, 1–10. https://doi.org/DOI: 10.22004/ag.econ.47866,

Bouma, J. A., Biggs, T. W., & Bouwer, L. M. (2011). The downstream externalities of harvesting rainwater in semi-arid watersheds: An Indian case study. *Agricultural Water Management*, *98*(7), 1162–1170. https://doi.org/https://doi.org/10.1016/j.agwat.2011.02.010

Mollinga, P. P. (2003). *On the waterfront: Water distribution, technology and agrarian change in a South Indian canal irrigation system*. Orient Blackswan.

R., L. M., & P., M.-S. (2005). Intensive Groundwater Use: Silent Revolution and Potential Source of Social Conflicts. *Journal of Water Resources Planning and Management*, *131*(5), 337–341. https://doi.org/10.1061/(ASCE)0733-9496(2005)131:5(337)

Wallach, B. (1984). Irrigation Developments in the Krishna Basin since 1947. *Geographical Review*, *74*(2), 127–144. https://doi.org/10.2307/214095

**Reviewer 3**

This paper is quite ambitious in attempting to deepen and enrichen the current state of the science around agent-based socio-hydrology modeling by reducing hydrological response units to individual farmholdings through the usage of a higher-resolution spatial scale. At the same time, it promises a fine-grain resolution on the large scale, but then explains that the agent behavior is simplified and that a large-scale model will be forthcoming in the future. For the most part, it is a valuable contribution that I enjoyed reading. However, I have a few questions and thoughts before publication.

*Dear reviewer, we thank you for the kind words. Below we suggest improvements to the manuscript based on the reviewer's comments in italic.*

1. It is unclear why the author has rendered a single farm into multiple HRUs when its holding expands beyond single grid cells. Does the inability to maintain one HRU per farm not detract from the representation goals of the study? Perhaps it does not, but an explanation of this . (Lines 85-86)

*In this study the main aim is to simulate farms as independently operated environments, ensuring that management options which are chosen by a farmer do not affect other farms. However, due to the incompatibility of a farm map and a gridded hydrological model due to the much higher resolution of the former, it is necessary to split some of these HRUs. However, these splitted HRUs are still owned by a single farmer and thus management decisions (e.g., switching crop type) by a farmer affect all HRUs and thus the entire farm they own.*

*We hope this explanation is clear and will include an explanation along these lines in an updated version of the manuscript.*

2. It is cautioned that the study is merely to showcase the model, but this feels dissatisfying somehow. A model is only interesting insofar as it is useful, and the reader needs more support beyond the vague notion that the authors enjoy hypothetical scenarios (Lines 365-367). Why is it not realistic? Is it stylized or semi-stylized? Does this effect its generalizability?

*Based on the comment of the reviewer (as well as other reviewers) we suggest to replace the current scenario with a more realistic scenario along the following lines:*

   1. *Use the Indian Development Human Development Survey (IHDS), which presents highly detailed information on 41.554 households, including crop types for the different growing seasons, household size, household income, expenditure, irrigation techniques, farm size etc etc.*

   2. *From the Indian agricultural survey, we will collect farm characteristics (i.e., marginal distributions), including farm sizes and crop types at tehsil level (comparative to counties in the US)*

   3. *Use an adapted version of the iterative proportional fitting to create a synthetic farmer population using the micro-level data from the IHDS yet fits the marginal distributions of the tehsil level agricultural survey*

*Then based on this data, land use data derived from satellite imagery, and the distribution of farm sizes, we will distribute those heterogeneous farmers spatially, which now include heterogeneous characteristics (according to the IHDS), such as crop types for the different growing seasons, household size, household income, expenditure, irrigation techniques, farm size etc.*

*Then, using historic crop prices, as obtained from the Indian Agricultural Marketing Information System, in combination with simulated yield based on farm size, potential yield and the simulated ratio of actual to potential evapotranspiration, we will simulate individual farmer income. Combined with inflation-adjusted household expenditures (from IHDS data) and crop expenditures (based on data obtained from the Ministry of Agriculture and Farmers Wellfare) we can calculate disposable income.*

*Next, we can include farmer adaptation behaviour, specifically the construction of irrigation wells, in the model. To simulate this, farmers without an irrigation well, look to (farm-size adjusted) income of neighbouring farmers with similar crop types but with an irrigation well. If the income difference of the agent's farm compared to those surrounding farms is higher than the implementation and upkeep costs of an irrigation well, the farmer will implement an irrigation well.*

*This also allows us to derive more in-depth conclusions, based on the heterogeneity of these agents (in addition to the presentation of the model framework and potential future applications).*

3. The elaboration of the findings and the conclusion are both insufficient. The paper's findings end strongly with the description of Figure 10, but there's little elaboration on what it means. There is in other words, scanty "discussion" of the model results. What do they tell us, the reader, in socio-hydrological terms (an expansion on socio-hydrology in the literature review could help with this)? Conclusions could also show further generalizability and the future potential for studies like this.

Saying that a large-scale model will happen in the future both undermines the initial claims of the article and fails to examine the purpose of the present one.  In short, I think the paper ought to be revised in order to promise less  at the beginning and offer more at the end.

*We thank the reviewer for their comment, and in a future version of the manuscript we will include the scenario as suggested above, which can also provide the basis for a more elaborate discussion and conclusion on socio-hydrological components. We also will include an additional paragraph in the introduction on socio-hydrological/ agent-based models.*

**Reviewer 4**

The paper has received several critical comments already, which I agree with, and therefore do not want to repeat the same comments and crticisms here.

First of all this is an ambitious attempt at developing a large-scale agent based socio-hydrological model. I applaud the authors for embarking on this adventure.

*First of all, thank you for your kind words. Below we replied to your comments in italic.*

However, as the other reviewers are saying, the authors present a rather superficial and half-hearted attempt at building such a model. It comes across to me as a "proof of concept" type of approach to announce to the world they are developing this model, and to demonstrate they have the elements of such a model in hand. To qualify as a scientific journal article, what lessons have been learned from this exercise? The authors may want to think about this some more.

*We thank the reviewer for their comment. In line with the reviewers comment (and also other comments), we suggest to replace the current scenario with the following, which considers a more realistic scenario which includes farmer heterogeneity based on survey data and endogenous adaptation:*

1. *Use the Indian Development Human Development Survey (IHDS), which presents highly detailed information on 41.554 households, including crop types for the different growing seasons, household size, household income, expenditure, irrigation techniques, farm size etc etc.*

2. *From the Indian agricultural survey, we will collect farm characteristics (i.e., marginal distributions), including farm sizes and crop types at tehsil level (comparative to counties in the US)*

3. *Use an adapted version of the iterative proportional fitting to create a synthetic farmer population using the micro-level data from the IHDS yet fits the marginal distributions of the tehsil level agricultural survey*

*Then based on this data, land use data derived from satellite imagery, and the distribution of farm sizes, we will distribute those heterogeneous farmers spatially, which now include heterogeneous characteristics (according to the IHDS), such as crop types for the different growing seasons, household size, household income, expenditure, irrigation techniques, farm size etc.*

*Then, using historic crop prices, as obtained from the Indian Agricultural Marketing Information System, in combination with simulated yield based on farm size, potential yield and the simulated ratio of actual to potential evapotranspiration, we will simulate individual farmer income. Combined with inflation-adjusted household expenditures (from IHDS data) and crop expenditures (based on data obtained from the Ministry of Agriculture and Farmers Wellfare) we can calculate disposable income.*

*Next, we can include farmer adaptation behaviour, specifically the construction of irrigation wells, in the model. To simulate this, farmers without an irrigation well, look to (farm-size adjusted) income of neighbouring farmers with similar crop types but with an irrigation well. If the income difference of the agent's farm compared to those surrounding farms is higher than the implementation and upkeep costs of an irrigation well, the farmer will implement an irrigation well.*

*This also allows us to derive more in-depth conclusions, based on the heterogeneity of these agents (in addition to the presentation of the model framework and potential future applications).*

My second point is that the paper does not articulate for me a vision or underlying design of such an agent based socio-hydrological model? Of course there are agent based models developed at small scales. What are the kinds of questions that the authors want to answer using this larger-scale model? I especially want them to think of "large" scale. How do they organize the model, the agents, the interactions, feedbacks etc in such a a way as to answer these questions? At present, the model focuses only on the mechanics of building the model.

*The feedbacks across long-ranges are included in the model through the hydrological component. It is unlikely that a farmer downstream at the delta of the Krishna river is directly influenced (for example through a social network) by a farmer upstream near the Western Ghats. However, when the farmer upstream applies irrigation water to their land which then (partly) evaporates, this water is not available for the downstream farmer. The main aim of this model is to allow the investigation of basins as a whole, because effects of behaviour are felt throughout the basin. Therefore, the immediate interactions (in the scenario proposed above) are between farmer agents are local, yet the compounded effect is basin-wide.*

*To address the reviewers' comment, we first suggest a paragraph in the introduction to specify some of the effects that can be experienced at long ranges:*

> *Moreover, the hydrological system includes many connections across large scales, and consider farmer heterogeneity. For example, farmers at the head-end of a command area have access to a much larger and reliable water supply than tail enders* (Mollinga, 2003). *These incentivizes head-end farmers to adopt water-intensive high-return crops, reducing water availability downstream* (Wallach, 1984). *Similarly, upstream famers that invest in rainwater harvesting techniques reduce the amount of water available downstream* (Bouma et al., 2011). *Yet another example is through groundwater use, where individual well users lower the groundwater table in the entire region* (R. & P., 2005). *And while some farmers might be able to invest in deeper wells, other farmers are left behind driving them further into poverty* (Batchelor et al., 2003).

*Moreover, we will try to dig deeper into some of these aspects following the storyline suggested above, adding some visualizations and discussion on these points.*

Finally, one of the features of agent based models from standard water management models is the idea of emergent dynamics or patterns that arise from the two-way feedbacks between humans and nature (water, hydrology), and between different agents and different kinds of agents. I am concerned that the way the model is presented (perhaps this is an issue of presentation quality) that this model comes across as just a water management model, and the two-way feedbacks is missing and the interactions between different agents is either not present or does not lead to emergent dynamics. I would like the authors to think through this and improve the presentation of the model.

*We thank the reviewer for their comment, and hope to address this partly by the scenario presented above, and partly by including the following figure, to specify more clearly the two-way feedback between the hydrological and human components (in addition to the feedbacks on farmer crop management and reservoir management shown in Figure 2).*

[Figure]

Given the journal, I do not consider this a traditional scientific article. Yet, I would like them to substantially improve the presentation to make it more interesting and appealing to the readers. I recommend major revision, but the paper should ultimately be published in GMD

*We thank the reviewer for their comment and hope that we can use the suggestions above to improve the manuscript for publishing in GMD.*

Batchelor, C. H., Rama Mohan Rao, M. S., & Manohar Rao, S. (2003). Watershed development: A solution to water shortages in semi-arid India or part of the problem? *Land Use and Water Resources Research*, *3*, 1–10. https://doi.org/DOI: 10.22004/ag.econ.47866,

Bouma, J. A., Biggs, T. W., & Bouwer, L. M. (2011). The downstream externalities of harvesting rainwater in semi-arid watersheds: An Indian case study. *Agricultural Water Management*, *98*(7), 1162–1170. https://doi.org/https://doi.org/10.1016/j.agwat.2011.02.010

Mollinga, P. P. (2003). *On the waterfront: Water distribution, technology and agrarian change in a South Indian canal irrigation system*. Orient Blackswan.

R., L. M., & P., M.-S. (2005). Intensive Groundwater Use: Silent Revolution and Potential Source of Social Conflicts. *Journal of Water Resources Planning and Management*, *131*(5), 337–341. https://doi.org/10.1061/(ASCE)0733-9496(2005)131:5(337)

Wallach, B. (1984). Irrigation Developments in the Krishna Basin since 1947. *Geographical Review*, *74*(2), 127–144. https://doi.org/10.2307/214095

---

## Author Response (AR2)

Dear reviewer,

Thank you for the very useful suggestions. We have adapted the abstract to include more specifics about the behaviour, added information about the soils to the case study description (section 3), included an short explanation of the equity issues in the case study descript and the conclusions.

Finally, we incorporated all textual edits.